

# Genome-wide identification of long non-coding (lncRNA) in *Nilaparvata lugens*'s adaptability to resistant rice

Wenjun Zha[1], Sanhe Li[1], Huashan Xu[1], Junxiao Chen[1], Kai Liu[1], Peide Li[1], Kai Liu[1], Guocai Yang[1], Zhijun Chen[1], Shaojie Shi[1], Lei Zhou[1] and Aiqing You[1,2]

[1] Hubei Key Laboratory of Food Crop Germplasm and Genetic Improvement, Food Crops Institute, Hubei Academy of Agricultural Sciences, Wuhan, China
[2] Hubei Hongshan Laboratory, Wuhan, Hubei, China

## ABSTRACT

**Background:** The brown planthopper (BPH), *Nilaparvata lugens* (Stål), is a very destructive pest that poses a major threat to rice plants worldwide. BPH and rice have developed complex feeding and defense strategies in the long-term co-evolution.
**Methods:** To explore the molecular mechanism of BPH's adaptation to resistant rice varieties, the lncRNA expression profiles of two virulent BPH populations were analyzed. The RNA-seq method was used to obtain the lncRNA expression data in TN1 and YHY15.
**Results:** In total, 3,112 highly reliable lncRNAs in TN1 and YHY15 were identified. Compared to the expression profiles between TN1 and YHY15, 157 differentially expressed lncRNAs, and 675 differentially expressed mRNAs were identified. Further analysis of the possible regulation relationships between differentially expressed lncRNAs and differentially expressed mRNAs, identified three pair antisense targets, nine pair *cis*-regulation targets, and 3,972 pair co-expressed targets. Function enriched found arginine and proline metabolism, glutathione metabolism, and carbon metabolism categories may significantly affect the adaptability in BPH when it is exposed to susceptible and resistant rice varieties. Altogether, it provided scientific data for the study of lncRNA regulation of brown planthopper resistance to rice. These results are helpful in the development of new control strategies for host defense against BPH and breeding rice for high yield.

# INTRODUCTION

Long non-coding RNAs (lncRNAs) are a class of non-protein-encoding RNAs longer than 200 bp and have little or no evidence for coding capability (*Wang et al., 2014*; *Zhang et al., 2014*). LncRNAs can be further classified into long intergenic non-coding RNAs (lincRNAs), natural antisense transcripts, and intronic RNAs (incRNAs) (*Dogini et al., 2014*; *Heo, Lee & Sung, 2013*; *Zhang & Chen, 2013*). Functional analyses of lncRNAs have indicated that they are potent *cis*- and *trans*-regulators of gene transcription. Presently, lncRNAs have been identified in insects, including *Drosophila melanogaster* (*Quinn et al.,*

Corresponding authors
Lei Zhou, yutian_zhou83@163.com
Aiqing You, aq_you@163.com

2016), *Anopheles gambiae* (*Jenkins, Waterhouse & Muskavitch, 2015*), *Tribolium castaneum* (*Yang et al., 2021*), *Leptinotarsa decemlineata* (*Wan et al., 2013*), *Nilaparvata lugens* (*Xiao et al., 2015*) and so on. Previous studies results showed many lncRNAs were related to immunity and metabolism (*Valanne et al., 2019*), mediate resistance (*Feng et al., 2020*), and so on. A series of research results show that insect lncRNAs are not only affected by the use of insecticide (*Liu et al., 2017*; *Meng et al., 2021*; *Shi et al., 2022*; *Zhu et al., 2021*; *Zhu et al., 2017*), but also induced after eating Bt-resistant transgenic crops (*Lawrie et al., 2021*). These processes may lead to the evolution of insect defense systems to adapt to plant resistant genes or insecticides. However, lncRNAs are less conserved even among evolutionarily related species; therefore, insects may also exhibit poor conservation features.

The brown planthopper, *N. lugens* Stål (abbreviated as BPH hereafter), is a phloem-feeding insect of cultivated rice *Oryza sativa* and many wild *Oryza* species (*Xue et al., 2014*). BPH damages rice growth and spreads plant viruses, including rice-ragged stunt virus and rice grassy stunt virus, which leads to a large decline in rice yields (*Cabauatan, Cabunagan & Choi, 2009*). BPH completes its life cycle in 23–32 d, so it often complete 3–12 generations per year in the tropics and temperate areas (*Zheng, Zhu & He, 2021*). Rice is a staple food for the Chinese population and has been cultivated in the lower Yangtze Valley in China for about 10,000 years, and the BPH may have shifted from *Leersia* to *Oryza* about 0.25 million years ago. Through co-evolution, BPH has strongly adapted to host rice (*Sezer & Butlin, 1998*; *Zheng, Zhu & He, 2021*).

In long-term co-evolution, rice has developed complicated defense systems against BPH. During this process, through habitat location and host location, acceptance, suitability, and regulation phases, some BPH populations have emerged to overcome the resistance of these rice varieties' resistance by carrying a major resistance gene (*Cheng, Zhu & He, 2013*). Although insecticides have been widely used to control damage from BPH and other pests, overuse of them has led to resistant BPH resurgence and has caused environmental problems that threaten human health (*Senthil-Nathan et al., 2009*). The first resistant rice variety against BPH was discovered as early as 1969, over 40 BPH resistance genes have been reported now (*Akanksha et al., 2019*; *Li et al., 2019*). The first two resistant genes were designated *Bph1* and *Bph2*; since then, the subsequent resistance genes have been named *Bph3–40*. In 40, 20 genes (*Bph1–9, Bph17, Bph19, Bph25–26, Bph28, Bph30–33*, and *Bph37–38*) have been identified in indica varieties (*Balachiranjeevi et al., 2019*; *Hu et al., 2018*; *Jing et al., 2017*; *Prahalada et al., 2017*; *Wang et al., 2018*; *Yang et al., 2019*), whereas the other 20 genes (*Bph10–16, Bph18, Bph20–24, Bph27, bph29, Bph34–36*, and *bph39–40*) are from wild species of rice (*Akanksha et al., 2019*; *Jing et al., 2017*; *Kumar et al., 2018*; *Li et al., 2019*; *Zhang et al., 2020*). However, it was found that while resistance genes resist BPH, they also accelerate changes in the physiology and behavior of BPH, such as prolonging the developmental period and decreasing reproductive yield (*Du et al., 2009*; *Nguyen et al., 2019*; *Senthil-Nathan et al., 2009*). BPH that feed on resistant rice for a long time may slowly evolve into a new and virulent BPH population to overcome rice resistance (*Peng et al., 2017*).
To clarify the molecular mechanism of co-evolution between BPH and resistant rice, in our previous report, proteomics and miRNA sequencing were performed using a susceptible rice variety (TN1) as a control and a moderately resistant rice variety (YHY15) carrying the resistance gene *BPH15* (*Zha & You, 2020*; *Zha et al., 2016*). However, the molecular mechanism and regulatory network are still unclear. Many studies have proofs lncRNAs involved in key biological processes, including cell differentiation (*Ganegoda et al., 2015*), transcription regulation (*Kurokawa, 2011*), dosage compensation (*Quinn et al., 2016*), and so on.

The BPH varieties TN1 and YHY15 were further used for lncRNA expression profile analysis in this study. The new lncRNAs from RNA-seq datasets were identified and the lncRNAs expression level in the two BPH varieties were compared. The differential expression lncRNAs were further used for screening the differentially expressed target genes. These results will provide a basis for us to further understand the co-evolutionary molecular mechanism of rice planthopper and provide a reference for high yield and pest control of rice.

## MATERIALS AND METHODS

### Plants and insects

For this study, two virulent BPH populations were full sib-mated for at least 40 generations and fed with TN1 and YHY15 plants in a climate chamber under a 16-h light/8-h dark cycle at 25 ± 1 °C and 70% relative humidity at the Hubei Academy of Agricultural Sciences, Wuhan, China. Twenty insects of the two virulent BPH populations (hereinafter referred to as TN1 and YHY15) were obtained, respectively. Whole insects of the two samples were immediately frozen in liquid nitrogen and stored at −70 °C until RNA isolation.

### Construction of lncRNA sequencing library and RNA sequencing

Total RNA was extracted from brown planthopper using a Trizol kit (Invitrogen, Carlsbad, CA, USA) and further purified with the RNeasy kit (Qiagen, Hilden, Germany). The purity and quality of RNA were determined using Nanodrop instrument. Total RNA (1-μg) from TN1 or YHY15 samples (three biological replicates for each treatment: a total of six samples) was taken for library construction. The ribosomal RNA was removed to retain all coding RNA and ncRNA. The first-strand of cDNA was synthesized using random primers. Then, the second-strand cDNA was synthesized using dNTPs (dUTP instead of dTTP), RNase H and DNA polymerase I and was purified by QIAquick PCR Purification Kit. After end repair, poly(A) and sequencing adaptor addition, the cDNA was digested by UNG (Uracil-N-Glycosylase) enzyme. The fragments with expected size were selected by agarose gel electrophoresis. Finally, the constructed libraries were sequenced with Illumina Hiseq 4000.

### lncRNA prediction and new transcription analysis

The raw data were processed using filtering adaptors, containing more than 50% of low quality (Q-value ≤ 20) bases, and trimming the reads whose number of N bases accounted

for more than 10% of the total by fastp (version 0.18.0) (*Chen et al., 2018*). The reference BPH genome and the annotation files were downloaded from the National Center for Biotechnology Information database with the accession NO. GCF_014356525.1 or InsectBase (http://insect-genome.com/planthoppers/) (*Ma et al., 2021*). The rRNA mapped reads were removed after short reads mapping to the ribosome RNA (rRNA) database by Bowtie2 (version 2.2.8) (*Langmead & Salzberg, 2012*). The remaining reads were further mapped to the reference genome using HISAT2 (version 2.1.0) with "-rna-strandness RF" and other parameters set as a default (*Kim, Langmead & Salzberg, 2015*). The transcript reconstruction was performed using software Stringtie (version 1.3.4) and HISAT2 (*Kim, Langmead & Salzberg, 2015*; *Pertea et al., 2016*). Two softwares, CNCI (version 2) and CPC (version 0.9-r2) (http://cpc.cbi.pku.edu.cn/) were used to assess the protein-coding potential of novel transcripts by default parameters (*Kong et al., 2007*; *Sun et al., 2013*). The intersection of both non-protein-coding possible results were chosen as long non-coding RNAs.

## Differential expression genes screening and function enrichment analysis

Software StringTie quantified transcript abundances in a reference-based approach. A FPKM (fragment per kilobase of transcript per million mapped reads) for each transcription region was calculated to quantify its expression abundance and variations, using RSEM software (*Li & Dewey, 2011*). RNAs and lncRNAs differential expression analysis were performed using DESeq2 software between two groups (*Anders & Huber, 2010*; *Love, Huber & Anders, 2014*). The genes/transcripts with the parameter of FDR below 0.05 and absolute fold change ≥1.5 were considered differentially expressed genes/transcripts. The statistical power of this experimental design, calculated in PROPER is 0.72 (*Wu, Wang & Wu, 2015*). Differentially expressed coding RNAs were then subjected to enrichment analysis of GO functions and KEGG pathways (*Boyle et al., 2004*; *Kanehisa et al., 2008*). A Protein-Protein interaction network was identified using String v10 and visualized using Cytoscape software (v3.7.1) (*Shannon et al., 2003*; *Szklarczyk et al., 2015*).

## lncRNA-mRNA association analysis

The interaction between antisense lncRNA and mRNA was predicted by the software RNAplex (version 0.2) (http://www.tbi.univie.ac.at/RNA/RNAplex.1.html) (*Tafer & Hofacker, 2008*). lncRNAs in less than 10-kb up/downstream of a gene were identified to *cis*-regulators, and the lncRNA co-expressed with protein-coding genes were considered to have *trans*-regulation function. The target genes were further used for GO and KEGG enrichment analysis.

## Data validation by qRT-PCR

Total RNA was extracted using RNAzol® RT RNA Isolation Reagent (GeneCopoeia, Rockville, MD, USA). Total RNA from each sample was reverse transcribed in a 25-μL reaction using Surescript™ First-Strand cDNA Synthesis Kit (GeneCopoeia, Rockville,

**Table 1 Data statistics.**

| Sample | Clean_reads | Mapped to rRNA (%) rRNA | Unmapped_Reads (%) | Unmapped (%) RNA | Unique_Mapped (%) | Multiple_Mapped (%) | Total_Mapped (%) |
|---|---|---|---|---|---|---|---|
| TN1-1 | 94,416,426 | 21,089,196 (22.34%) | 73,327,230 (77.66%) | 14,622,000 (19.94%) | 52,349,637 (71.39%) | 6,355,593 (8.67%) | 58,705,230 (80.06%) |
| TN1-2 | 81,139,172 | 13,502,638 (16.64%) | 67,636,534 (83.36%) | 19,586,598 (28.96%) | 42,284,619 (62.52%) | 5,765,317 (8.52%) | 48,049,936 (71.04%) |
| TN1-3 | 80,883,132 | 4,825,924 (5.97%) | 76,057,208 (94.03%) | 21,373,172 (28.10%) | 48,362,573 (63.59%) | 6,321,463 (8.31%) | 54,684,036 (71.90%) |
| YHY15-1 | 73,618,386 | 3,705,790 (5.03%) | 69,912,596 (94.97%) | 13,773,065 (19.70%) | 49,954,636 (71.45%) | 6,184,895 (8.85%) | 56,139,531 (80.30%) |
| YHY15-2 | 86,328,074 | 4,098,928 (4.75%) | 82,229,146 (95.25%) | 46,978,467 (57.13%) | 30,536,210 (37.14%) | 4,714,469 (5.73%) | 35,250,679 (42.87%) |
| YHY15-3 | 89,676,312 | 10,973,130 (12.24%) | 78,703,182 (87.76%) | 21,125,300 (26.84%) | 50,326,027 (63.94%) | 7,251,855 (9.21%) | 57,577,882 (73.16%) |

MD, USA). The sequences of the primers used are indicated in Table S1. The *NlRPS11* gene of BPH was used as an internal control gene. qRT-PCR was performed using a BlazeTaq™ SYBR® Green qPCR mix2.0 protocol (GeneCopoeia, Rockville, MD, USA). The 20-μL reaction volume consisted of forward and reversed primers (1-μL), BlazeTaq™ SYBR® Green qPCR mix (10-μL), ddH2O (6-μL), cDNA (2-μL). The selected genes were verified using an ABI 7900HT Fast Real-Time PCR System with a cycling temperature of 60 °C and a single peak on the melting curve to ensure a single product. Each sample's relative transcript levels were obtained using the $2^{-\Delta\Delta Ct}$ method (*Rao et al., 2013*). At least three replicates were tested per sample. The obtained data were subjected to unpaired a two-tailed Student's *t*-tests using GraphPad Prism software (version 8). Different stars were used to denote the significant variations at the $P < 0.05$ level. All the numerical data in figures are presented as means ± standard deviation (SD) of three independent replications.

## RESULTS

### High-throughput sequencing of BPH lncRNAs

To investigate the dynamic variation of lncRNAs in BPH when fed with TN1 (susceptible rice variety) and YHY15 (moderately resistant rice variety), the whole-transcriptome strand-specific RNA sequencing for two BPHs was conducted using three biological replicates. In total, after removing low-quality reads, more than 506 million clean reads (above 99.85% raw reads) were generated with Q30 value in all libraries exceeding 92.95% though high-throughput sequencing (Table 1). After removing the reads mapped to rRNA, the mapping rate of clean reads against the BPH reference genome ranged from 42.87–80.30%. About 60% of the reads were mapped to the exon region, and about 20% were mapped to the intron and intergenic regions, respectively (Table S2).

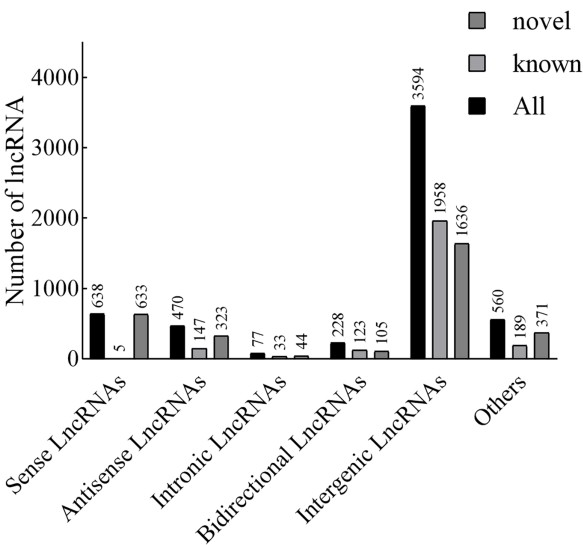

**Figure 1 The lncRNA type statistics.**

## Identification of lncRNAs in BPH

In this study, after the known coding mRNAs (transcripts and their splices) were filtered out from the mapped reads, largely expressed transcripts with length ≥200 bp and exon number ≥2 were selected. The results indicated that 10,937 potential lncRNA transcripts were identified. Then, these sequences were further processed using algorithms CPC2 and CNCI to assure the nonexistence of protein-coding domains. At last, 3,112 highly reliable lncRNAs were identified (Fig. S1).

According to the position of the new lncRNAs relative to protein-coding genes on the genome, new lncRNAs can be divided into five categories. In our results, the lncRNAs included 638 sense lncRNAs, 470 antisense lncRNAs, 77 intronic lncRNAs, 228 bidirectional lncRNAs, 3,594 intergenic lncRNAs, and 560 others (Fig. 1). Additionally, in terms of FPKM, the transcriptional abundance of lncRNAs was significantly lower than that of mRNAs, but there was no significant difference between TN1 and YHY15 (Fig. S2).

## Analysis of differentially expressed lncRNAs in BPH

To analyze the difference in expression of lncRNAs between TN1 and YHY15, the normalized expression of lncRNAs in two BPH were compared. In this study, the false discovery rate (FDR) below 0.05 and absolute fold change ≥1.5 were used to identify expression lncRNAs and mRNA differentially (Figs. 2A, 2B). A total of 157 differentially expressed lncRNAs (including 84 upregulation and 73 downregulation), and 675 differentially expressed mRNAs (including 281 upregulation and 394 downregulation) were identified (Fig. 2C), respectively.

## Function of differentially expressed lncRNAs

To explore the potential ncRNAs functions, lncRNA and mRNA association analysis was performed in three ways: base complementary pairing of lncRNA and mRNA (antisense), lncRNA's location in 10 kb upstream and downstream of its adjacent protein-coding genes

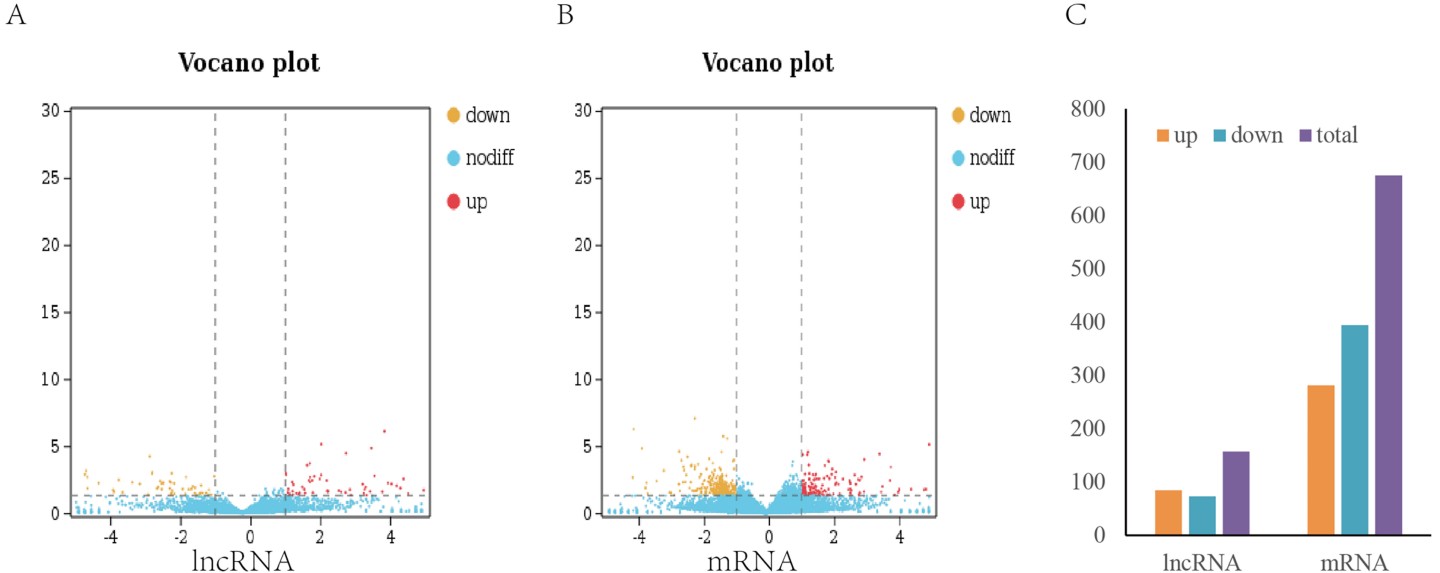

**Figure 2 Differentially expressed lncRNAs and mRNA in two BPH.** (A) The volcano plot of lncRNA. (B) The volcano plot of mRNA. (C) The number of differentially expressed lncRNAs and mRNA.

(cis-regulation), and lncRNA's correlation with co-expressed protein-coding genes identified predictable targets (co-expression). As the results showed, three antisense targets (Table S3), nine pair cis-regulation targets (include eight lncRNAs and eight mRNAs) (Table S4), and 3,972 pair co-expressed targets (include 156 lncRNAs and 643 mRNAs) were identified (Table S5).

For the three antisense targets, the expression profile of the lncRNAs and corresponding target genes indicated a similar trend. The expression levels of lncRNA MSTRG.12888.1 and it predicted that target ncbi_111044590 (homology of Dus4l, dihydrouridine synthase 4) was higher in TN1 than in YHY15, MSTRG.26118.1, and MSTRG.26199.1 with their target genes ncbi_111053346 (pilin gene-inverting like protein) and ncbi_120355505 (homology of TrwC, DNA relaxase/conjugal transfer nickase-helicase) showed lower expression level in TN1 than in YHY15 (Fig. 3A).

For the nine pair cis-regulation lncRNAs and targets, except lncRNA XR_005571984.1 and target gene ncbi_111050469 (zinc finger CCCH domain-containing protein) exhibited opposite expression trend (lncRNA XR_005571984.1 were higher expressed in TN1 than in YHY15, but mRNA ncbi_111050469 were lower expressed in TN1 than in YHY15), the other lncRNAs and their targets exhibited similar expression trend (expression trend were simultaneous upregulation or downregulation) (Fig. 3B). Ncbi_111058532 coding a serine/threonine-protein kinase, ncbi_111053346 coding a pilin gene-inverting like protein, ncbi_111053362 coding a F-box DNA helicase 1 (Fbh1), ncbi_111061470 is a homology of jerky protein, which facilitates Wnt signalling.

The function annotation was further performed for the 3,972 pair co-expressed targets, including inc 156 lncRNAs and 643 mRNAs (Fig. 3C, Table S6). First, Gene Ontology (GO) analysis was conducted to categorize these protein-coding genes. There were 23 classes of biological processes and these protein-coding genes were mainly enriched in
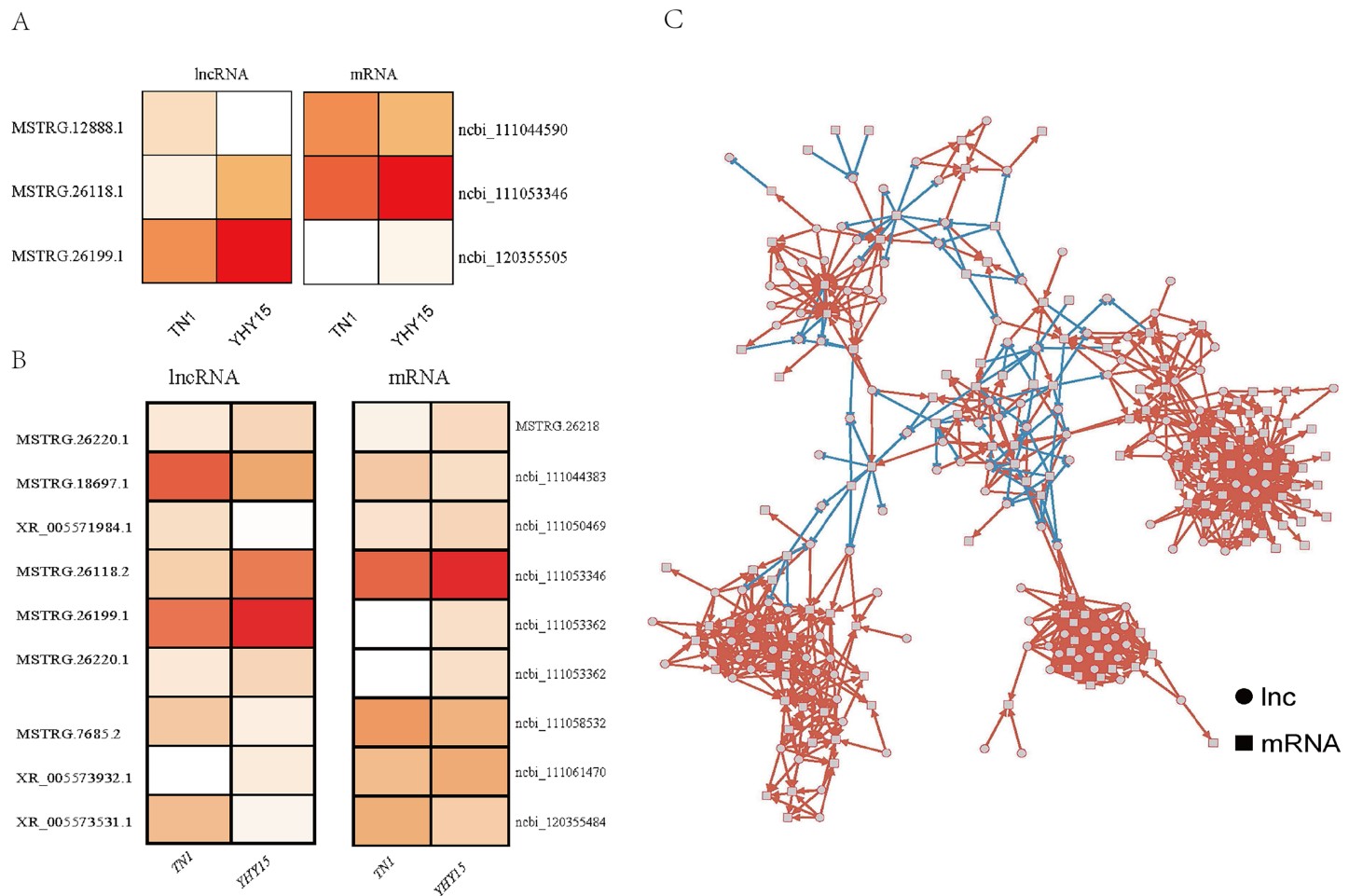

**Figure 3 The expression profile and regulatory networks of lncRNA and corresponding target genes.** (A) Antisense. (B) Cis-regulation. (C) Co-expression target regulatory networks, circle mean lncRNAs, square indicated mRNAs, red line means positive regulation, and blue line means negative regulation.

"single-organism process", "cellular process", and "metabolic process". Moreover, some essential stress resistance genes were identified as lncRNAs targets, including "response to stimulus", "immune system process", and so on (Fig. 4). These stress resistance genes homology may mainly related to the MAPK signalling pathway, including upstream regulation genes such as mitochondrial uncoupling protein 2 (ncbi_111047800), peroxiredoxin 1 (ncbi_111050813), toll-like receptor (ncbi_111064393), superoxide dismutase (ncbi_120349329), or downstream regulation genes such as defensin-like (ncbi_111045303), glutaredoxin-3 (ncbi_111048799), transferrin (ncbi_111050388), nemo-like protein (ncbi_111044309), serine/threonine-protein kinase (ncbi_111045417, ncbi_111048296, ncbi_111055048), P450 (ncbi_120356101) and so on (Table S7). These findings proposed that these differentially expressed lncRNAs might be involved in different biological processes, including stress resistance by regulating the expression of related protein-coding genes—significantly enriched cellular components, including cell,

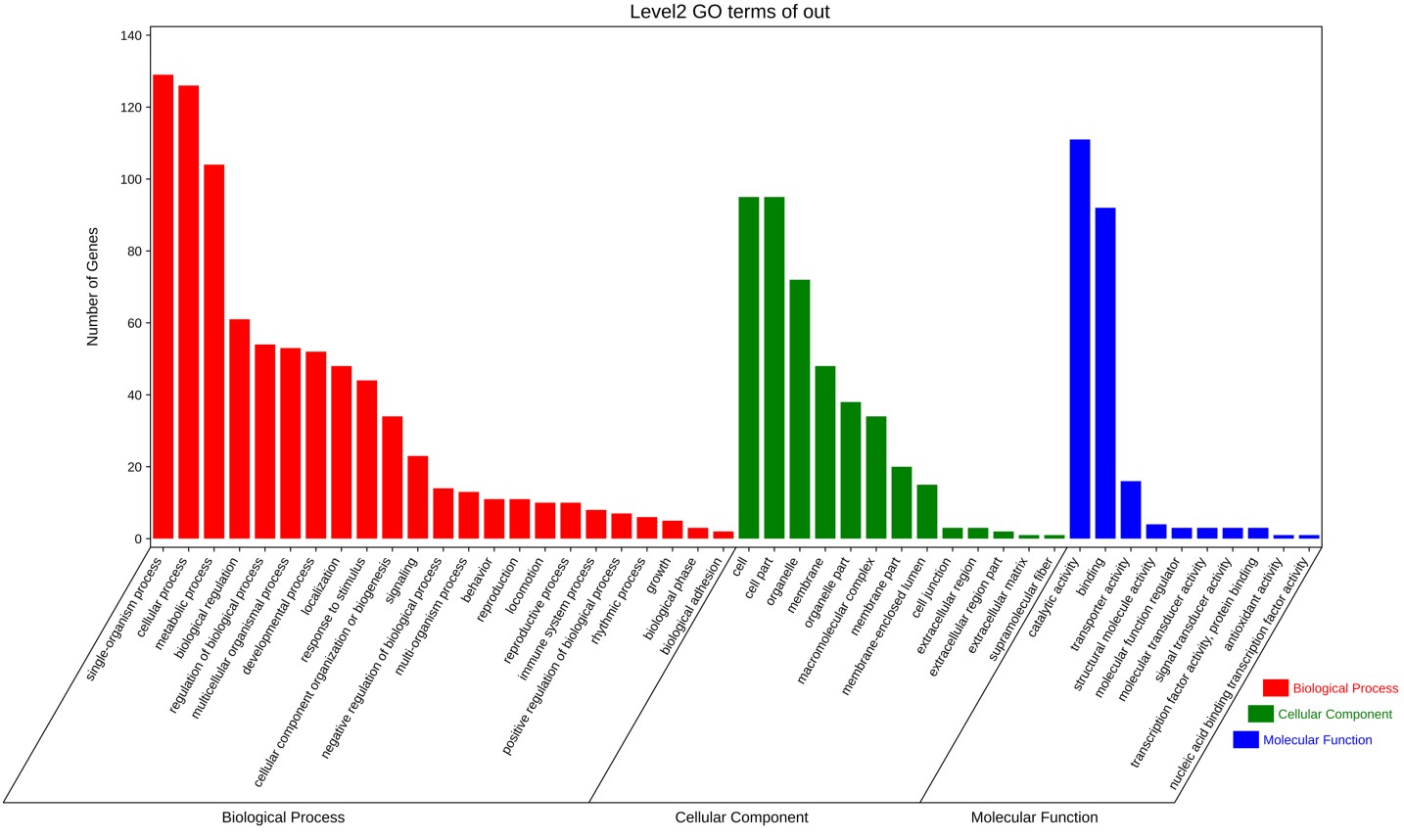

**Figure 4 Gene ontology analysis of co-expressed protein-coding genes with the differentially expressed lncRNAs.**

cell part, organelle, and membrane. The differential expression targets were mainly enriched in catalytic activity and binding in molecular function categories.

Moreover, based on the Kyoto Encyclopedia of Genes and Genomes (KEGG) analysis, these co-expression target genes were mainly enriched in metabolism, environmental information processing, organismal systems, cellular processes related pathways (Fig. 5, Table S8). Among these pathways, a part of categories enriched many genes, including metabolic pathways, signal transduction, transport and catabolism, and endocrine system. In metabolic-related pathways, arginine and proline metabolism, glutathione metabolism, and carbon metabolism categories were enriched 10 genes (Table S8). These findings suggested that lncRNAs have a significant effect on the adaptability of BPH when exposed to susceptible and resistant rice varieties.

## Validation of differentially expressed lncRNAs and mRNA in BPH

qRT-PCR was used to confirm the expression profiles of the RNA-seq data in two BPH. The results indicated that among 10 selected different expression mRNAs and lncRNAs, about 80% of mRNA expression trends were consistent with RNA-seq data, except ncbi_111060469 and ncbi_111061470 (Figs. 6A and 6B). The qRT-PCR results of lncRNA indicated that, except MSTRG.26199.1, MSTRG.26220.1, and

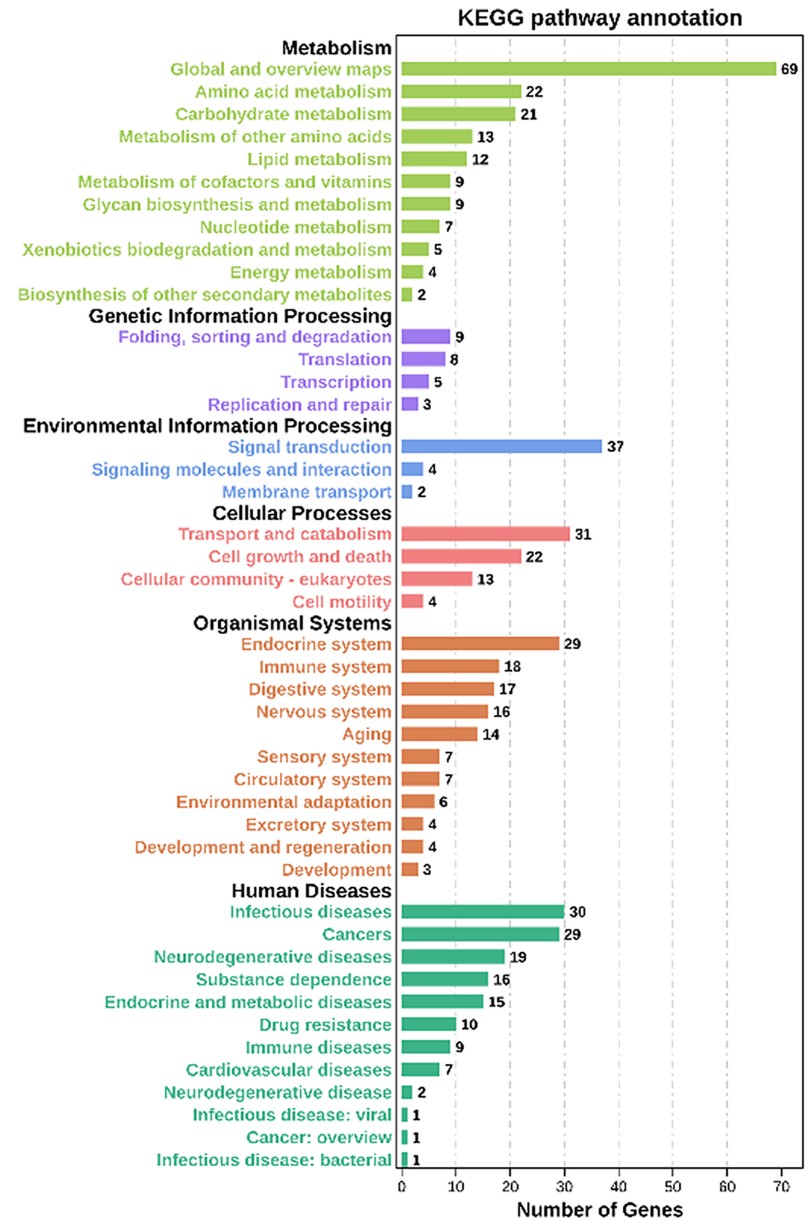

**Figure 5 KEGG analysis of co-expressed protein-coding genes with the differentially expressed lncRNAs.**

XR_005573932.1, expression trends of the other seven lncRNAs were consistent with RNA-seq and qRT-PCR, the gene number ratios attained 70% (Figs. 7A and 7B).

## DISCUSSION

The brown planthopper is a destructive pest that poses a significant threat worldwide rice cultivation. To explore the molecular mechanism of BPH adaptation to resistant rice variety. In our previous study, proteomic profiles and miRNA expression profiles of BPH were analyzed (*Zha & You, 2020*; *Zha et al., 2016*). In these studies, there has been evidence that miRNA may be involved in the co-evolution process of rice planthopper resistance,

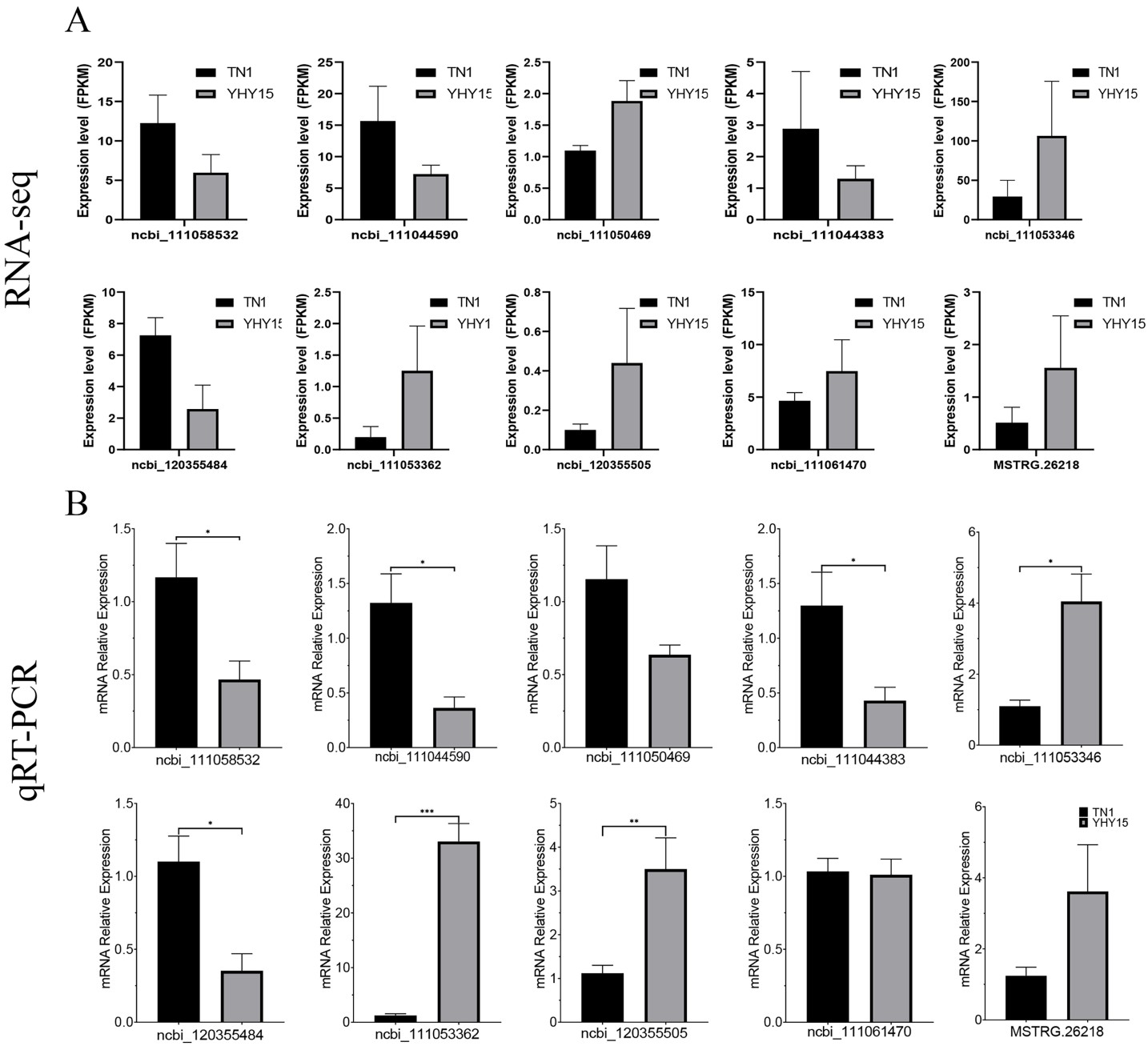

**Figure 6 Expression pattern of the selected target genes in BPH.** (A) Gene expression data for RNA-seq. Values are means ± SD of three technical replicates. (B) The qRT-PCR analysis of gene expression data. Error bars represent SEM for three independent experiments. Student's *t*-test: *P < 0.05, **P < 0.01, ***P < 0.001.

and proteome analysis also found many differentially expressed proteins. LncRNA is widely involved in plants and animals' development and life cycle regulation. In this study, the role of lncRNA in the co-evolution of rice planthopper and rice resistance will be further explored. As indicated in the results, 3,112 lncRNAs, which were slightly higher than the number obtained by *Xiao et al. (2015)* in the fat body, salivary gland, and antenna tissues of rice BPH was identified. In the initial stage, 10,937 potential lncRNA transcripts

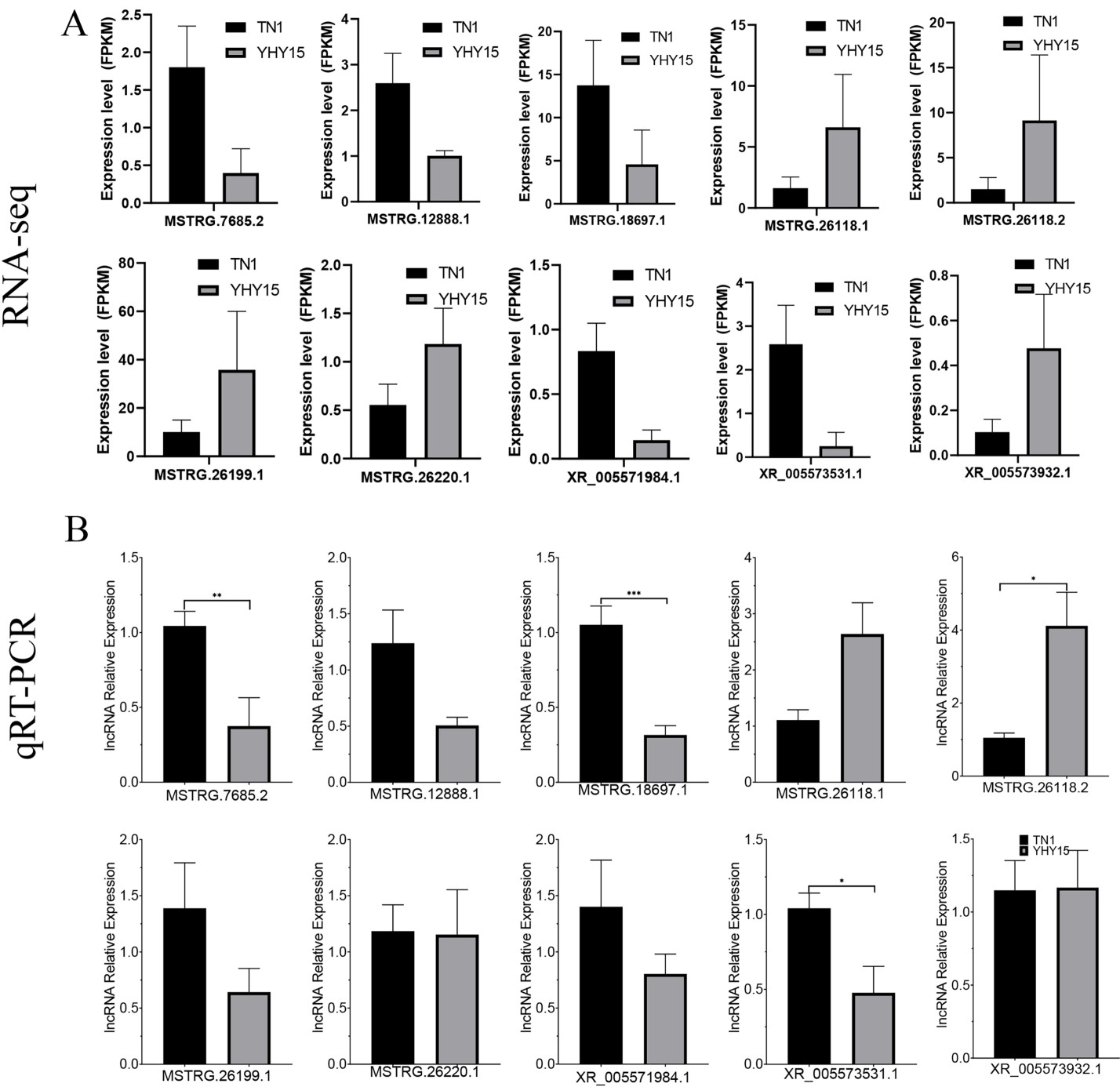

**Figure 7 Expression pattern of the selected lncRNAs in BPH.** (A) lncRNAs expression data for RNA-seq. Values are means ± SD of three technical replicates. (B) The qRT-PCR analysis of lncRNAs expression data. Error bars represent SEM for three independent experiments. Student's *t*-test: *$P < 0.05$, **$P < 0.01$, ***$P < 0.001$.

were identified, but the amount is much greater than that in most of the insects (*Jenkins, Waterhouse & Muskavitch, 2015*; *Valanne et al., 2019*). The CPC2 and CNCI were further used to assure the nonexistence of protein-coding domains. Then, 3,112 lncRNAs were identified for further analysis.

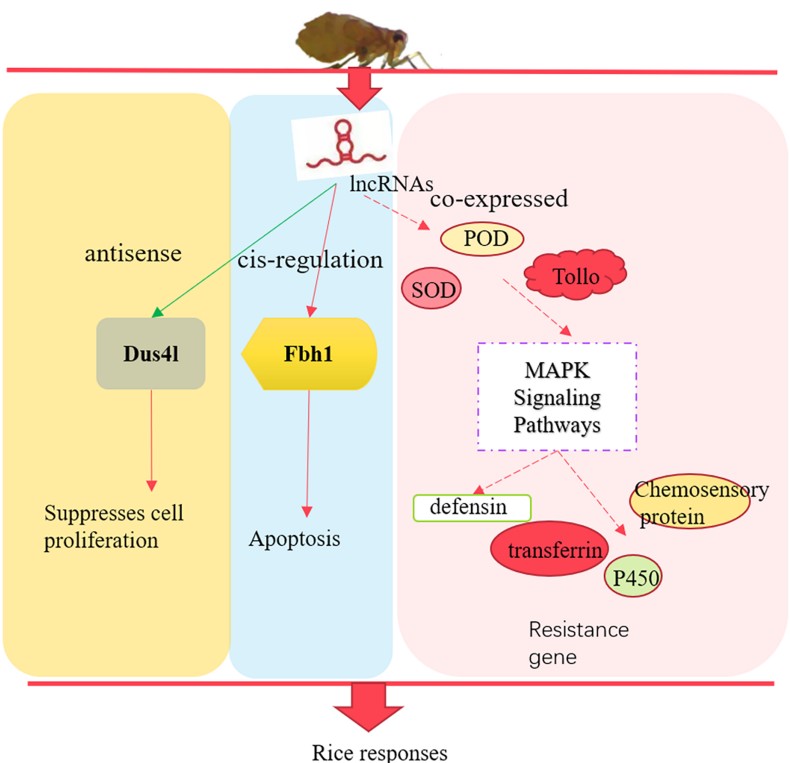

**Figure 8 The preliminary model of rice immune responses to BPH.**

It has been reported that the function of lncRNA is primarily to regulate the expression of coding genes. Therefore, the focus is on the analysis of differentially expressed lncRNA and differentially expressed mRNA in the two ecotypes of TN1 and YHY15, and association analysis was conducted in three possible regulatory relationships. To antisense targets, three pair was identified, ncbi_111044590 is a homology of *dihydrouridine synthase 4 like* (*Dus4l*) gene, which silently suppresses cell proliferation and promotes apoptosis in humans (*Li et al., 2020*). In BPH, ncbi_111044590 is downregulated in YHY15, it may be that YHY15 consumed resistant rice, and its cell growth was inhibited. To the nine pair *cis*-regulation targets, the target ncbi_111053362 is a homology of the *F-box DNA helicase 1-like* (*Fbh1*) gene. According to the report, FBH1 promotes DNA double-strand breakage and apoptosis in response to DNA replication stress and is important to restore normal mitotic progression (*Fugger et al., 2013*; *Jeong et al., 2013*; *Laulier et al., 2010*). Ncbi_111058532 is a serine/threonine protein kinase.
The mitogen-activated protein kinase (MAPK) family comprises serine/threonine kinases that mediate intracellular signaling. MAPK is reported in whitefly and diamondback moths to enhance insect resistance to pesticides and *Bacillus thuringiensis* toxin action, respectively (*Guo et al., 2020*; *Yang et al., 2020*).

To co-expression targets, the results of the enrichment analysis of the KEGG, in lysosome, glycolysis/gluconeogenesis have enriched nine genes, which is similar to the results of our previous miRNA and proteomic studies (*Zha & You, 2020*; *Zha et al., 2016*).

Additionally, glutathione metabolism, MAPK signaling pathway, and arginine and proline metabolism are also enriched in 9–10 genes, respectively. These three metabolic pathways are closely related to the resistance of insects (*Cen et al., 2020*; *Kostal et al., 2016*; *Raza et al., 2020*). Among these enriched genes, ncbi_120356101 is annotated as a P450 homologous gene. Presently, there have been reports that the P450 gene of rice BPH can promote adaptation to rice resistance (*Peng et al., 2017*). For stress resistance lncRNA target genes (Table S7), at least 17 genes may be directly or indirectly involved in MAPK signaling pathway. Mitochondrial uncoupling protein 2 (*Shao et al., 2017*), peroxiredoxin 1 (*Pang et al., 2021*), superoxide dismutase may be play upstream regulation genes roles, and the downstream genes may including defensin-like (*Jiang et al., 2011*), glutaredoxin-3 (*Zhang et al., 2016*), transferrin (*Brummett, Kanost & Gorman, 2017*), P450 (*Yang et al., 2020*) and so on. Additionally, the MAPK pathway activates transcription factors cAMP-response element-binding protein leads to P450-mediate imidacloprid resistance in whitefly (*Yang et al., 2020*). The targets gene P450 was also screened out in our previous studies. It further increases the correlation between these target genes and BPH adaptation to rice resistance.

## CONCLUSIONS

In summary, we identified 3,112 lncRNAs in susceptible (TN1) and resistant rice planthopper varieties (YHY15). After differentially expressed analysis between the two varieties, 157 differentially expressed lncRNAs and 675 differentially expressed mRNAs were screened out. The differentially expressed lncRNAs through their antisense, *cis*-regulation, and co-expressed targets involved in the adaptability of BPH to rice resistance (Fig. 8). Altogether, these results provide insights into the molecular mechanisms of BPH adaptability to resistant rice, which is essential for breeding rice and high yield.

## ACKNOWLEDGEMENTS

We are grateful to Dr. Minshan Sun and the Henan Assist Research Biotechnology Co., Ltd (Zhengzhou, China) for assisting in the sequencing and bioinformatics analysis. We would like to thank TopEdit for the English language editing of this manuscript.

### Funding

This work was funded by the National Key Research and Development Program of China (No. 2021YFD1401100), and the National Natural Science Foundation of China (No. 31501654). The funders had no role in study design, data collection and analysis, decision to publish, or preparation of the manuscript.

## Grant Disclosures

The following grant information was disclosed by the authors:
National Key Research and Development Program of China: 2021YFD1401100.
National Natural Science Foundation of China: 31501654.

## Competing Interests

The authors declare that they have no competing interests.

## Author Contributions

- Wenjun Zha conceived and designed the experiments, performed the experiments, analyzed the data, prepared figures and/or tables, authored or reviewed drafts of the article, and approved the final draft.
- Sanhe Li analyzed the data, authored or reviewed drafts of the article, and approved the final draft.
- Huashan Xu performed the experiments, authored or reviewed drafts of the article, and approved the final draft.
- Junxiao Chen analyzed the data, prepared figures and/or tables, and approved the final draft.
- Kai Liu performed the experiments, prepared figures and/or tables, and approved the final draft.
- Peide Li conceived and designed the experiments, authored or reviewed drafts of the article, and approved the final draft.
- Kai Liu analyzed the data, prepared figures and/or tables, and approved the final draft.
- Guocai Yang performed the experiments, prepared figures and/or tables, and approved the final draft.
- Zhijun Chen analyzed the data, authored or reviewed drafts of the article, and approved the final draft.
- Shaojie Shi performed the experiments, authored or reviewed drafts of the article, and approved the final draft.
- Lei Zhou analyzed the data, prepared figures and/or tables, and approved the final draft.
- Aiqing You conceived and designed the experiments, authored or reviewed drafts of the article, and approved the final draft.

## Data Availability

   The data is available at NCBI: PRJNA797862.

## Supplemental Information

Supplemental information for this article can be found online at http://dx.doi.org/10.7717/peerj.13587#supplemental-information.

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
