# Peer review of "Genome-wide identification of long non-coding (lncRNA) in Nilaparvata lugens’s adaptability to resistant rice"

_PeerJ, doi:10.7717/peerj.13587_

## Round 0.1 · original submission · Major Revisions

In this study out of twenty insects of the two virulent N. lugens populations (TN1 and YHY15), only two insects were studied for identified LncRNAs for adaptability to resistant rice. No control population/ non-virulent sample has been used to compare the differential expression of LncRNAs. This is the main drawback of the study. Clear replicate information is also missing.

·

Basic reporting

Sentence Structuring has to be improved throughout the manuscript. Please pay attention to words that have been used in the manuscript. For instance, in line 49, Oryza should be italicized; in line 57, co-evolutionary should be co-evolution.

Line 58-59: Revise it to make sense. How come BPH can carry a resistance gene?

Experimental design

From where the BPH colonies were collected? And if the authors maintained them, then for how many generations? Did they check for the presence of rice-ragged stunt/grassy stunt virus in the BPH they used for the experiment? Please include the confirmatory results in the complementary file.

Validity of the findings

Results are interesting. However, some additional data, especially on the representation part can improve the manuscript. For instance, how many lncRNAs were shared in between the two libraries, and how many were exclusive to each? The authors can give a Venn diagram for the same.

Did the authors compare with a complete check (CK) library? It can show the basal lncRNAs which are there irrespective of the BPH interaction with rice.

Based on the findings of the study, I suggest the authors make a conceptual model of lncRNA-mediated rice resistance breakdown by BPH and include in discussion and also add it as a picture. This will be hugely helpful for the readers to understand and correlate the findings.

Reviewer 2 ·

Basic reporting

.

Experimental design

.

Validity of the findings

.

Additional comments

This manuscript by Wenjun Zha and his colleagues investigated the lncRNA expression profiles of two Nilaparvata lugens populations’ adaptability to susceptible and resistant rice by the RNA-seq method. The manuscript needs careful editing in English grammar.
However, several points should be addressed before acceptance.

Minor points:
Line 31, there should be spacing between “.” and “Altogether”,
Line 31-32, I suggest that the description at lines 31- 32 “this study proves that lncRNA may be involved in BPH’s adaptability to resistant rice.” could be changed to “it provided scientific data for the study of lncRNA regulation of brown planthopper resistance to rice.”
Line 43, deleted “-”,
Line 44, show should change to “showed”,
Line 45, there should be spacing between “resistance” and “(”,
Line 49, Oryza should be italic,
Line 50, I suggest that the description at lines 50-51 “rice-ragged stunt viruses and rice grassy stunt”, such as “rice-ragged stunt viruse and rice grassy stunt viruse”,
Line 52-53, it might be better to delete the description “less than four generations in”,
Line 54, there should be spacing between “,” and “0”,
Line 73 and 242, the brown planthopper was abbreviated as BPH hereafter. Why use “BPHs”?
Line 78, why did you choose “a moderately resistant rice variety (YHY15)”, Why not choose a high resistant variety?
Line 80, please modify the description “proof”, such as “a proof or proofs”,
Line 92, the brown planthopper was abbreviated as BPH hereafter, “N. lugens” should change to “BPH”,
Line 92, dose the brown planthopper have been reared on resistant varieties for a long time, or was it treated by the resistant varieties after several generations? Dose the physiology and behavior of BPH have changed?
Line 108-109, in the description, change the location of “size”,
Line 152, “®” should be superscript,
Line 180 and 186, lncRNA should be “lncRNAs”,
Line 189, deleted “identified”,
Line 193, LncRNA should change “lncRNA”,
Line 194, Mrna? It is “mRNA”,
Line 195, “cis-” should be italic,
Line 199, there are “-” between number and word (9-pair) in some paragraph, however, there are not “-” between number and word (9 pair) in some paragraph, please modify,
Line 243, there should be a spacing between “analyzed” and “(”,
Line 252, “insect” should change “insects”,
Line 253, lncRNA should be “lncRNA”,
Line 276, “P450” was not italic,
Line 277, “P450 gene” need not to be italic,
Line 278, “The” should be “the”,
Line 279, “cAMP-response element-binding protein” need not to be italic,
In the introduction, it needs more detail about the function of lncRNA.

Reviewer 3 ·

Basic reporting

There are some typos and grammatical errors throughout the manuscript. Some examples,
Line 48: Nilaparvata lugens -> N. lugens.
Line 49: Oryza species -> Oryza (italics)

Line 241: worldwide rice plants -> worldwide rice cultivation.

Experimental design

I wonder on what basis only 10 genes were selected for qRT-PCR.
Author could use same RNA isolation method for both RNAseq and qRT-PCR. Any special reason for choosing different protocols.

Validity of the findings

From the overall results and discussion, we are not able to get which candidate genes are involved in the pest resistance. Apart from general GO and KEGG, key responsible genes information could be needed for readers to understand necessity to carried out this kind of works and future experimental design to be conduct on this area of research.
I would like to suggest author to emphasis significant differences and key regulatory genes/networks in both results and discussion.

---

## Round 0.2 · accepted · Accept

I am happy to accept the manuscript for publication in PeerJ.

·

Basic reporting

The authors have addressed my queries satisfactorily.

Experimental design

The authors have addressed my queries satisfactorily.

Validity of the findings

The authors have addressed my queries satisfactorily.

Additional comments

The authors have addressed my queries satisfactorily.

Reviewer 2 ·

Basic reporting

The author has revised the manuscript completely according to the reviewer’s suggestions; therefore, I recommend the revised version of the manuscript can be accepted and published.

Experimental design

GOOD

Validity of the findings

GOOD